# Physical and Mechanical Properties of Rapeseed Straw Concrete

**DOI:** 10.3390/ma15238611

**Published:** 2022-12-02

**Authors:** Maya Hajj Obeid, Omar Douzane, Lorena Freitas Dutra, Geoffrey Promis, Boubker Laidoudi, Florent Bordet, Thierry Langlet

**Affiliations:** 1Innovative Technologies Laboratory (LTI), University of Picardie Jules Verne, Avenue des Facultés, CEDEX 1, 80025 Amiens, France; 2CODEM, 56 Rue André Durouchez, 80080 Amiens, France

**Keywords:** rapeseed concrete, compressive strength, thermal conductivity, biomass materials, lightweight construction, plant particles

## Abstract

This paper investigates an innovative building material based on rapeseed concrete. This material is a non-load-bearing insulating concrete, which is intended for use in the construction of wood-frame walls thanks to its thermophysical properties. It is composed of particles of rapeseed straw, lime, and cement. First, this work proposes to characterize rapeseed straw aggregates according to the place of cultivation, the year of harvest, and the size of the straw strands. For this purpose, straws of three different origins and different years of harvest were chosen. Aggregate sizes of 10 mm and 20 mm in length were selected. In a second step, this study focuses on the effect of the type of rapeseed straw aggregates on the mechanical resistance and thermal conductivity of bio-based concrete. The results obtained showed that the influence of the different parameters on the compressive strength was stronger than that on the thermal conductivity. On the one hand, rapeseed concrete made with 10 mm straw exhibited a lower thermal conductivity, averaging at 0.073 W.m^−1^.K^−1^. On the other hand, concrete manufactured with the 20 mm size aggregates demonstrated a higher mechanical strength, which remained relatively low and closer to 0.22 MPa. Finally, 20 mm-long aggregates offered the best compromise between mechanical and thermal resistance.

## 1. Introduction

Nowadays, the sector of building and residential is responsible for 45% of energy consumption in France, as well as 26% of emissions of global warming gases [1]. The use of bio-based materials in the construction sector as insulating and supporting materials appears to be an important solution: they have low thermal conductivity, relatively high specific capacity, and can regulate temperature and humidity in enclosed spaces, which leads to the reduction of building energy consumption [2,3,4,5].

Bio-based materials can be obtained from a variety of biomass sources: oilseeds (rapeseed, etc.), starch (maize, wheat, etc.) and sugar (beet, etc.), fibers (flax, hemp), forestry resources, herbaceous plants, or organic industrial by-products. They are particularly interesting because they can capture and store CO_2_ from the atmosphere during the photosynthesis process [6]. In France, different sectors are involved in the development of bio-based materials, traditionally hemp, flax, and wood. In addition to absorbing carbon dioxide during plant growth, bio-based concrete also absorbs carbon dioxide through the carbonation of lime or associated binders. These types of concrete present excellent hygric performance [7,8,9,10] and they contribute to improving the acoustic comfort of houses [11,12,13,14].

They are also good insulators [11,15,16,17], for instance, the thermal conductivity of hemp concrete ranges from 0.100 W.m^−^^1^.K^−^^1^ to 0.236 W.m^−^^1^.K^−^^1^ for bio-based concrete densities between 400 kg/m^3^ and 887 kg/m^3^ [18,19]. The thermal properties depend directly on many aspects: the density of the materials, the relative humidity of the ambient, the ratio of mixed aggregates, and the water content of the material [20,21]. 

Other factors play an important role in the properties of bio-based materials, such as type of binders, aggregate size, and possible pre-treatment, compaction energy, and molding and manufacturing methods [22,23,24]. Thanks to the plant aggregates, with their high porosity and of course their low density, bio-based concretes are generally light and, through their insulating power, contribute to reducing energy consumption. 

Recently, rapeseed was presented as an interesting bio-based product for the construction sector since it currently occupies 12% of the cultivated land in France and represents 25% of European rapeseed production [25,26]. Another critical factor is its mode of transportation and its impact on the carbon footprint [27,28,29]. Moreover, this plant is cultivated every year, which ensures a regular, local, and fast supply. After processing, the plant can provide powders, fibers, or aggregates, used in bulk as insulation or for the manufacture of bio-sourced concrete. 

Research conducted by Rahim [30] compared rapeseed concrete and hemp concrete, showing that rapeseed concrete presented better hygric performance. In addition, Rahim [31] found a lower thermal conductivity for rapeseed concrete (0.299 W.m^−1^.k^−1^) than hemp concrete (0.355 W.m^−1^.k^−1^). Therefore, taking into account the potential of rapeseed concrete, the aim of this study is to further enhance its thermal performance so that it can be utilized as an insulator while also possessing mechanical properties that enable it to be employed as a self-supporting wall.

However, it is known that one of the most common causes that affect the thermomechanical characteristics of bio-based concrete is the interaction between the binder and the vegetable straw [32]. Thus, this study has chosen to analyze the effect of the straw’s variety on concrete properties. For this purpose, straws from different regions in France and from different harvest years were submitted to a large physical, chemical, and morphological characterization campaign, and consequently, the composition of rapeseed concrete was optimized. 

Finally, an innovative non-load-bearing insulating concrete containing rapeseed straw, lime, and cement was developed. In addition, its thermomechanical properties, including compressive strength and heat conductivity, were determined.

## 2. Materials and Methods

CODEM’s facility and laboratory tools were used for material fabrication and testing for this study.

### 2.1. Rapeseed Straws

Rapeseed straws were harvested in two French regions, Somme and Marne, during the 2015 and 2016 crop years. The grinding was carried out by external service providers using two milling techniques: shearing and impact. Two sizes of straw were chosen to produce concrete specimens: 0.25–10 mm and 0.25–20 mm. These sizes were targeted since it was verified that straws smaller than 10 mm do not provide satisfactory mechanical performance [33]. Aggregate batches were referred to by the name and year of the harvested region, for instance: M15 corresponds to straw grown in the Marne region and harvested in 2015; similarly, for M16 (Marne 2016) and S16 (Somme 2016). The grinding size completes the information, for example, M16-10 for a crushing size of 10 mm. All samples were milled by shearing and an additional sample was ground by impact. The combination of these different parameters resulted in a total of 7 batches of rapeseed straws, as shown in Table 1 and illustrated by Figure 1.

#### 2.1.1. Microstructural Characterization

Scanning Electron Microscopy (SEM) was used to examine the microstructure of rapeseed straws. This technology uses electron–matter interactions to produce very high-resolution 3D images of a sample surface in the nanoscale range. The microstructural analysis was carried out by employing a scanning electron microscope (Vega3, Tescan) from Codem’s laboratory, which was equipped with an energy-dispersive X-ray spectroscope (EDS). The polished thin sections were gold coated in the Quorum Q150R ES rotary-pumped coating system. The images were obtained using secondary electron (SE) detector, operated at a 10.0 kV accelerating voltage for morphology visualization.

In this study, straws were observed longitudinally and transversely, as well as glued with carbon tape to avoid any change due to electron beams.

#### 2.1.2. Methods for Chemical Characterization

Plant fibers are composed mainly of hemicelluloses, cellulose, lignin and solubles. Nevertheless, many molecules such as sugars, proteins, lipids, tannins, and mineral matter are present in plant fibers. The Van Soest method [34,35] was used to determine chemical composition or the quantity of various molecular components on rapeseed straw. For this purpose, a sample was divided into three bags and treated with a neutral detergent solution (NDF) and washed. This is a substitution test with multiple chemical detergents that was performed on the samples in several steps. The parallel treatment of the three bags allowed for measuring by elimination the quantity of residual organic matter at the end of each extraction step. All the steps are diagrammed in Figure 2.

Currently, there is a standard (AFNOR XP U44-162) [36] allowing for the characterization of organic matter by biochemical fractionation on organic amendments. Another former AFNOR standard, NF V18-122 [37], specifies a technique of treating biomass with neutral and acid detergents, as well as sulfuric acid. 

**Figure 2 materials-15-08611-f002:**
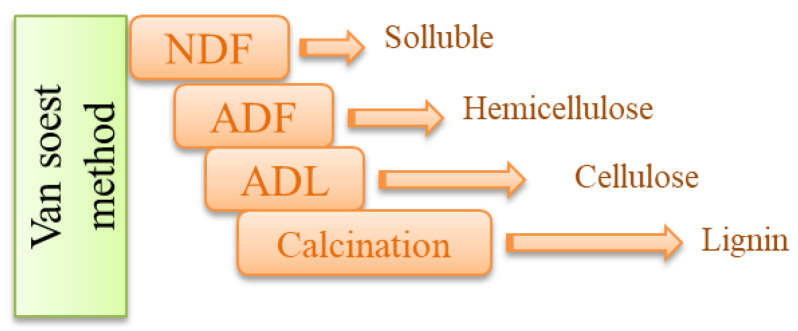
Schematic drawing of the Van Soest method for biomass decomposition [38,39].

#### 2.1.3. Methods for Physical Characterization

For a successful formulation of rapeseed concrete, it is necessary to first understand the physical characteristics of the rapeseed straws. In this study, the density, distribution of aggregate sizes, water absorption, and moisture content of all types of rape straws are of particular interest. All of these tests are carried out as recommended by the Rilem Technical Committee [40]. After being collected from the fields, the rapeseed straw bales were stored for six months in laboratory. Then, the straws were crushed and placed in large bags according to their type and size. The purpose of this exploratory study is to identify the properties of untreated rapeseed straws without conditioning, as well as after conditioning at 23 °C and 50%RH.

##### Bulk Density

The apparent density of the straws, after crushing and without treatment, was measured with a cylindrical container with a volume of five liters. The measurement was repeated ten times to reduce the degree of uncertainty. The test consisted of filling the known-volume cylinder with rapeseed straw in three portions. The level is adjusted without settling. Straw mass was calculated by subtracting the mass of the empty cylinder. The bulk density was determined by dividing the average mass of ten trials by the total volume.

##### Particle Size Distribution

The granular distribution of the rapeseed straw was carried out with the sieving method. To homogenize the straw and achieve a good granular distribution, a sampling method was required. It consisted of mixing the rape straw, taking a quantity, spreading it in a circle, and then dividing it into 4 portions. To obtain the required quantity, it is important to select two diagonal portions and then repeat the homogenization process.

The sample weighs approximately 80 to 100 g. In this study, sieves with mesh sizes of 8, 6.3, 4, 3.15, 2, 1, 0.50, 0.25, 0.125, 0.08, 0.063, and 0.01 mm were utilized. The experiment is performed with a mechanical sieve for thirty minutes. Verifying that the initial mass is equal to the sum of the masses retained in each sieve was crucial to this task. The passing mass must not surpass 2% of the total mass of the rape straw [41]. Figure 3 depicts the results of the sieving residues

##### Water Absorption

The amount of water absorbed by the plant aggregates gives an idea of the amount of pre-molding water required for concrete manufacturing. The water absorption of aggregates was measured with the coffee pot method recommended by Rilem [40]. This technique consists of weighing 20 g of dry or conditioned rapeseed straws at 23 °C and 50%RH and then dipping them in a coffee pot containing 500 mL of water under light pressure from a piston (Figure 4). The measurement of the mass of rapeseed straw was carried out outside the pot, after total immersion in water for 1 h. The ratio of absorbed water is calculated using the initial mass of rapeseed straw [40].

##### Moisture Content

The moisture content is the amount of water vapor adsorbed in the material’s pores at room temperature. In this study, the moisture content was determined by drying three samples at 70 °C and then conditioning them at 23 °C/50%RH. When the mass of samples remained stable for at least three consecutive days, equilibrium was reached. Finally, the moisture content was computed by subtracting the mass in the equilibrium state at 50%RH from the mass in the dry state.

### 2.2. Rapeseed Concrete

Rapeseed concrete, known as BINP, is the study’s target material. Therefore, samples were prepared at CODEM by mixing straws with pre-molding water for 2 min. The pre-molding water was calculated based on the results of water absorbed by the rapeseed straw, and after that, the binders were added. The mixture remains dry for a period of 2 min and then water was added to hydrate the binders. The total mixing time is 10 min.

All formulations were made with the same volume percentages, i.e., 80 ± 10% for rapeseed straw. The binder used in this study is a mixture of cement and hydraulic lime with a percentage of 15 ± 10%. The mass ratio of water and binder was W/B = 0.9.

Following formulation preparation, samples were stored in a curing chamber at 23 °C and 50%RH. According to the application, samples were molded into two distinct shapes. Figure 5a represents 15 cm × 15 cm × 15 cm mechanical test specimens, while Figure 5b depicts 25 cm × 25 cm × 8 cm thermal conductivity test specimens.

#### 2.2.1. Thermal Conductivity Test

The guarded hot plate was used to measure the thermal conductivity (λ) of rapeseed concrete specimens. The tests were performed on samples measuring 25 cm × 25 cm × 8 cm (Figure 5b) in accordance with NF EN 12667 and NF EN 12664 [42].

Three samples with different densities were prepared for each composition. Thermal conductivity test was performed after 60 days a dry state.

After 30 days of curing, samples were pre-dried in an oven at 70 °C, weighed until the mass was stable and then placed in a 25 °C chamber until equilibrium was reached. In order to perform the measurements at various temperatures, the hot and cold plates (top and bottom sides) were adjusted so that all samples have an average temperature of 10 °C. A thermocouple was put between the overlapping surfaces of the two samples to eliminate contact resistance. Thermal conductivity was measured perpendicular to the compacting direction on the samples.

#### 2.2.2. Mechanical Test

The mechanical properties of rapeseed concrete were investigated using the hydraulic servo-controlled compression testing machine (Figure 6) at a displacement rate of 0.02 mm/s according to NF EN 12390-2 [43] and NF EN 196-1 [44]. Tests on 15 cm × 15 cm × 15 cm samples (Figure 5a) were carried out after 28 days of curing. In this investigation, the mechanical tests are measured parallel to the direction of settlement. Four cubic samples per formulation are prepared for the compressive strength test with a settlement control based on the initial density of the samples (1200 kg/m^3^). 

## 3. Results and Discussion

### 3.1. Rapeseed Straws

#### 3.1.1. Microstructural Characterization

Plant straws are generally the stems of plants (rape, hemp, flax, etc.). They are composed of several layers: the inner part (center) can be filled, as in corn straw, and is known as the pith. Other types of plants, such as hemp and wheat straw have a hollow space [45,46]. Rapeseed straw is the first type and, unlike hemp, has a pith in the center, which is formed of small porous bags. The diameter of this internal part varies depending on the plant’s size and maturity, as well as the variety. For rapeseed straw, the diameter of the pith ranges between 50 and 100 μm [38]. The epidermis is the plant’s protective layer and the most rigid part of the plant stem. Finally, there is the fibrous bundle, which is a unitary set of fibers that ensures the transport of water and proteins to other parts of the plant between the center and the epidermis. According to the literature, it has three layers. 

Figure 7 illustrates different parts of the rapeseed straws from the Somme and Marne. A cross-section of Somme straw harvested in 2016 allowed the identification of the pith (Pith) (Figure 7a), tracheid (trA) and vessels (Va) that act as water transporters (Figure 7b). The fiber bundle constitutes the phloem (PhL) (Figure 7c), vascular cambium and the xylem (Xyl), which is the set of vessels and tracheid, the vascular cambium responsible for radial growth of the stem. The pith (Pith) with its vessels in a longitudinal section of rapeseed straw from Marne 2015 can be seen in Figure 7d–f, as well as the epidermis (Epi), the most rigid part that covers the straw structure (Figure 7d,f), composed of the phloem (PhL) and the xylem (Xyl).

Finally, the SEM micrographs of the Marne 2016 aggregates (Figure 7g–i) showed mold (Moist) traces corresponding to the lighter spots on the snapshots. Damage to the pores and structure of the straw can be observed. Certainly, mold has spread throughout the straw as a result of inappropriate storage conditions.

#### 3.1.2. Chemical Characterization

Plant fibers are made up of several types of chemical molecules. Cellulose composes the rigid part of the plant stem while hemicellulose and lignin ensure the bond between the cellulose molecules 

The proportion of these components in each plant varies based on several factors, including climate, harvest year, type of growing soil, and plant size. Table 2 presents the chemical compositions of various types of vegetable aggregates as reported in the literature. In general, cellulose constitutes nearly half of the composition, while hemicellulose and lignin account for between 10–20% and 10%, respectively, with some exceptions.

Table 2 also shows the chemical compositions of the rapeseed straws obtained in this study, which are M16, S16, and M15. For the three batches, it is observed that the ratios of cellulose, hemicellulose, and lignin values are nearly identical. Nonetheless, a higher proportion of soluble fraction and less hemicellulose were observed when compared to those found in the literature. This is particularly true for the Marne 2016 straws, which contained a 29% soluble fraction. This can be explained by the presence of fungal growth in this batch, as seen previously. This increase in the soluble matter is accompanied by a drop of almost a third in the amount of hemicellulose, certainly due to its deterioration.

#### 3.1.3. Physical Characterization

##### Particle Size Analysis

When dealing with lignocellulosic stalks, square-mesh sieves are of limited interest because they do not take into account the elongation of the aggregates [51]. Nevertheless, a separate examination of the sieves after the test reveals that the average length of the particles changes little overall. This is presented in Figure 8, where only the particle thickness and width allow conditioning the passage from one sieve to the other.

From a comparative point of view, the influence of climate is more significant on rapeseed straws with diameters ranging from 0.25 to 20 mm. For example, it can be seen in Figure 8a that the percentage of particles on the 6.3 mm sieve is 35% for S16, whereas it is only 10% for M16. The majority of M16 0.25–20 mm rapeseed straws have a width of 1–2 mm, while the majority of S16 0.25–20 mm straws have a width of 2–4 mm. This difference is not observed for straws ranging from 0.25 to 10 mm (Figure 8b). In terms of the harvest year, the influence is more noticeable on straws ranging from 0.25 to 20 mm (Figure 8c,d).

The influence of straw cutting size on the particle size distribution of rapeseed aggregates is visualized in Figure 8e–g. Straws with a diameter of between 0.25 and 10 mm typically have a width between 0.5 and 2 mm, while those with a diameter between 0.25 and 20 mm have a width between 1 and 4 mm. Concerning the type of aggregate crushing, Figure 8h reveals that the granular distribution of the M16–20 mm straws is similar for the two types of crushing (impact and shearing).

The particle size distribution of each type of rapeseed straw is listed in Table 3. In total, 50% of the straws have a width of less than 1.5 mm, except for S16 straws, which exceed 2.25 mm, and M15 straws, which measure 2.02 mm. Regarding dust content, straws cut to 10 mm (M16-10, S16-10, and M15-10) exceed the critical limit of 2%, which may affect the interaction between the binder and the straw during concrete production [52]. This usually results in the poor setting of the concrete and a reduction in its mechanical performance.

##### Bulk Density

The bulk density of rapeseed straws is presented in Table 4. The range of values is between 90 and 104 kg/m^3^ for 10 mm aggregates and 65 and 75 kg/m^3^ for 20 mm aggregates.

The grinding process and harvest year had little effect on the density of rapeseed straws. Given that the density values of Somme straws were slightly higher than that of Marne straws, the climate had minimal effect on density. However, it is evident that aggregate size is the primary factor affecting bulk density.

Since the percentage of cellulose was higher in Somme 16 than in Marne 16 and Marne 15, the apparent density of the straws may be associated with their chemical composition. The lower density of the Marne straw and its lighter color (Figure 1) compared to the other batches confirm the presence of a greater quantity of pith, which is less dense than the fiber part.

In addition, it can be seen that bulk density was largely dependent on aggregate size, with differences for each batch (M16, S16 and M15) in the order of 20 to 32 kg/m^3^. Thus, the samples with the 10 mm straw had a higher density in the order from 30 to 45%. This is due to the better arrangement of the smaller straws, which reduces the voids between the aggregates.

The bulk and true specific densities, as well as the porosity of a variety of straws, are presented in Table 5. It is remarkable that apparent density varied significantly between studies. This difference is directly related to the granulometry of the aggregates for the same type of straw. The density of the aggregates studied by Viel [37] was inversely proportional to the size of the examined straws.

##### Water Absorption

Figure 9 illustrates the maximum water content of various types of straw after drying and conditioning at 23 °C and 50%RH. Comparing the water absorption capacities of different cultures of rapeseed straws of the same size (M16-10, S16-10, and M15-10 or M16-20, S16-20, and M15-10), no significant differences can be observed. However, there is a slightly higher absorption capacity for rapeseed straws of 0.25–10 mm compared to those of 0.25–20 mm. This may be due to the higher dust content in the smaller aggregate batches.

Due to their high porosity, the amount of water absorbed by plant aggregates generally varies from 200 to 400% of their initial mass. Comparatively, the absorption of wheat straw is in the range of 300–330% [56], and that of hemp between 250% and 360% [33,57,58].

The water absorption rates of rapeseed straws were approximately 400%. This greater absorption capacity compared to other types of straw is explained by the spongy texture of the pith in the heart of rapeseed straw, which is better suited to retain water.

##### Moisture Content

The moisture contents of raw rapeseed straws are described in Table 6. The type of shredding and the size of the aggregates after grinding do not seem to have a significant influence on the moisture content of the straw. The moisture content of raw straw differs according to harvest location and year. The percentage was greater for Marne 2016 rapeseed straws than for Marne 2015 and Somme 2016 rapeseed straws.

The moisture content of rapeseed straw used in a building must be less than 15% [59] and each batch of this work fulfills this requirement. The average moisture content of M15 and S16 was between 5 and 6%, which is a low value for plant-based materials, and Marne 16 had the highest value (close to the limit of 15%). It should also be noted that the climate and storage conditions have a direct effect on the moisture content of vegetable aggregates. As previously mentioned, this difference in the M16 is undoubtedly due to the development of mold in this batch.

### 3.2. Rapeseed Concrete

As rapeseed concrete is intended for use as a self-bearing insulator, this material has a low compressive strength in comparison to other building materials. The application of this material will be either in the form of blocks placed in a wooden frame, cast, or shattered rapeseed concrete. 

Considering that raw materials play an important role in the final material performances, a comparative study is carried out according to the different batches of rapeseed straw on two characteristics: the mechanical resistance in compression and the thermal conductivity. The formulation is the same for all samples and is described in paragraph 2.2. To take into account the different years of straw harvesting, an additional batch was created by mixing M15-20 and S16-20 aggregates in equal parts.

#### 3.2.1. Thermal Behavior of Rapeseed Concrete 

Bio-sourced materials present a good thermal performance thanks to their high porosity and low density. The purpose of this section is to understand the effect of density, aggregate size, the effect of climate and harvest year, and the type of grinding on thermal conductivity. 

##### Influence of Density on the Thermal Conductivity

Table 7 exhibits the thermal conductivity of rapeseed straws from two distinct origins and two harvest years, for particle sizes 0.25–10 mm and 0.25–20 mm, as well as the thermal conductivity of rapeseed concrete produced from impact-crushed straws. Twenty different samples were evaluated and the results were sorted by decreasing the order of concrete density. According to the findings, the thermal conductivity of rapeseed concrete ranged from 0.070 to 0.087 W.m^−1^.K^−1^. The properties are comparable to those of hemp concrete, making these materials effective insulators for construction.

As previously stated, thermal conductivity varies with density and this relationship was supported by a number of studies [20,23,24,60]. Figure 10 illustrates the distribution of thermal conductivity values as a function of sample density. For instance, M15-20 samples with a density of 348 kg/m^3^ have a thermal conductivity of 0.073 W.m^−1^.K^−1^, whereas samples with a density of 608 kg/m^3^ have a thermal conductivity of 0.083 W.m^−1^.K^−1^. This behavior is confirmed for all rapeseed concrete manufactured with all aggregate types.

Nonetheless, it should be highlighted that the presence of mold on the M16 straw influences the results, as concrete made with the M16 straw has a rather high thermal conductivity despite its relatively low density for all particle sizes.

Johra [61] explained that, for insulating materials used in building construction, porosity is responsible for the positive association between density and thermal conductivity in the range of densities above 100 kg/m^3^. Cerezo [20] also confirmed that the amount of voids, or the size and quantity of pores, accounts for the relationship between density and thermal conductivity; hence, an increase in density leads to a decrease in porosity, particularly the inter-aggregate pores.

##### Influence of Aggregate Size on the Thermal Conductivity

Another factor that can influence the thermal conductivity of concrete is the size of the aggregates. Considering the density of the samples, it can be seen in Table 7 that thermal conductivity increased with the size of the rapeseed straw aggregates. It must be pointed out that the same percentage by volume of material is used to make each sample, resulting in a greater mass of rapeseed straw for concretes with 0.25–10 mm aggregates than for those with 0.25–20 mm. As seen previously (Table 4), the density of 0.25–10 mm straw is higher than that of 0.25–20 mm straw due to the aggregate arrangement.

On the other hand, the distribution of small-sized aggregates results in a lower porosity, especially between aggregates, which reduces the conduction in the solid part of the matrix of materials. Moreover, Walker et al. [24] showed that while the thermal conductivity of hemp concrete is proportionate to the amount of straw used, the type of binder has no effect on its thermal conductivity.

The thermal conductivity is directly related to the orientation and rearrangement of the matrix [62]. The compaction pressure has a greater effect on samples manufactured with 20 mm straws than on samples made with 10 mm straws, which are more easily shaped. This compaction pressure alters the orientation of the fibers, which can promote elongation and produce approximately unidirectional fibers, hence enhancing the flow and thermal conduction. 

##### Influence of Grinding Type on the Thermal Conductivity

The effect of grinding type on thermal conductivity can be verified by comparing samples (with the same composition) ground by impact and shear techniques.

If a linear relationship between thermal conductivity and density is assumed for these samples, Figure 10 illustrates how, given the same density, M16 impact concrete had thermal conductivity values that were higher than M16 shear-ground concrete. In fact, impact grinding degrades the porosity of rapeseed straws more, hence diminishing their thermal characteristics.

However, the mold appeared to have an effect on the thermal conductivity of the rapeseed concrete, as the thermal conductivity values of the BINP concrete manufactured with M16 straws are especially high despite their low density. The development of mold in rapeseed straw M16 was detected in several experiments (microstructure, moisture content and the amount of soluble in chemical composition). Figure 11 illustrates the mold growth on samples of rapeseed concrete. Moldy straw is therefore not suggested for use in bio-sourced concrete, as it can modify the material’s properties. On top of that, it was also mentioned in the literature that mold can affect the hygrothermal performance of bio-sourced materials [63].

#### 3.2.2. Mechanical Behavior of Rapeseed Concrete 

Despite their good thermal performance, insulating materials in buildings generally have low mechanical strength. For this reason, the following sections present a study on the influence of several parameters (place and year of harvest of rapeseed straw, size of the aggregates, and type of crushing) on the compressive strength of a non-load-bearing insulating concrete (BINP).

##### Influence of Climate and Harvest Year 

As previously mentioned, the rapeseed straws used in this investigation were obtained during a two-year period (2015 and 2016) in two French areas (Marne and Somme). The influence of the climate and harvest year on straw properties can thus be seen in the mechanical performance of the concrete. Figure 12 exhibits BINP compressive strength as a function of crop year, growing region, and straw size.

For aggregates cut to 20 mm in length, the harvest year appeared to have little influence on the compressive strength of concrete since the M15-20 and M16-20 curves are very close. This is less obvious for a 10 mm-long straw considering the gap between the M15-10 and M16-10 curves.

Regarding the influence of the place of cultivation of rapeseed on the mechanical properties of concrete, it is observed that the samples made with straw from the Marne, whose climate is rather continental, have a higher ductility than the samples manufactured with straw from the Somme, which has a more oceanic climate (curves in blue and brown). The latter showed a more brittle behavior with a more pronounced failure, with a yield strength of about 0.18 MPa for S16-20, and 0.12 MPa for S16-10.

For the samples made with Marne straw, no distinction can be made between elastic and plastic behavior and no failure is observed. This behavior is of elastoplastic type with an increase in the stress due to the densification of the material. Over a certain load, the binder cracks and all the efforts are supported by the straw aggregates [64]. This phenomenon can be explained by the amount of pith in the Marne straw. This portion of the rapeseed plant has a sponge-like texture that appears to contribute to mechanical strain dampening, preventing rupture. Figure 1 shows that the Marne straw is lighter in color than the other batches, indicating the presence of a white pith.

This is not the case for the straw harvested in the Somme. Microscopic observation showed that the pith in the core of rapeseed straw from the Marne is greater than that from the Somme (Figure 7). This may be due to the climate or the nature of the cultivation soil.

##### Influence of Aggregate Size

Raw materials have a significant impact on final material performance. As a result, understanding the effect of rapeseed straw size on concrete mechanical performance is critical. Results showed that, for all harvest years and regions of straw, BINP compressive strength increased when using larger aggregates (Figure 12).

The same results were found in [65] on concrete at 28 days made with hemp particles for three different sizes of aggregates. Arnaud et al. [65] verified that concrete made with small aggregates was less porous than others and stated that the carbonation of the samples made with small-sized aggregates was not completely finished at 28 days. Similarly, investigations made in [66] confirmed that mineral cement with larger aggregates presented a higher compressive strength.

Additionally, it seems that 20 mm straws intertwine with each other forming a type of net and preventing the propagation of the crack more efficiently than a 10 mm straw.

The effect of the aggregates’ size can also be verified by the other tests made on the straws, especially on the absorption of water. Straws with a diameter of 10 mm absorb more water than those with a diameter of 20 mm. Regarding the dust content, the 10 mm straw size exceeds the recommended limit. Both of these phenomena play an important role in rapeseed concrete, as the dust content contains molecules that prevent the proper interaction between the aggregate and the binder, as well as higher water absorption, which can modify the hydration of the binder [32].

Finally, comparing the compression performance of concrete made with 10 mm and 20 mm rapeseed straw, it can be seen that there is a significant improvement for the smaller size, with a 50% increase in mechanical strength. However, the thermal performance is reduced. In fact, the thermal conductivity of rapeseed concrete made with 20 mm rapeseed straws is 13% lower than that of concrete made with 10 mm rapeseed straws.

##### Influence of the Grinding Type

Figure 13 exhibits the mechanical properties of rapeseed concrete produced using M16-20 aggregates and impact and shear grinding techniques. It was observed that the mechanical strength curves for both types of grinding have the same form, with no variation between the elastic and plastic regions, and no failure was detected.

However, samples produced with rapeseed straw crushed by impact grinding present a slight improvement in mechanical strength curves. Indeed, this grinding type damages more the porosity of rapeseed straws, which can favor the mechanical performance.

##### Case of Mixing Two Batches of Straw

As seen in Figure 12, concretes made with 20 mm straws exhibited superior mechanical properties. Furthermore, the origin of straws appeared to play a significant role in this behavior, for instance, Somme straws appeared to be more rigid due to their stiff stems while Marne straws were more flexible thanks to their bigger piths. Thus, combining both characteristics could lead to concrete with enhanced mechanical properties.

Figure 14 illustrates the compressive strength of a formulation made of a blend of 20 mm-diameter straw from Marne 2015 and Somme 2016. By combining the straw aggregates from the Marne and the Somme, the resulting concrete exhibited an intermediate behavior, with an elastic domain that is more pronounced than with the M15-20 mm aggregates alone and an elastoplastic domain that is more extensive than for S16-20 mm aggregates. Additionally, compared to a single-type straw batch, the blend increased the concrete’s compressive strength by 10% and mechanical strength to a maximum of 0.22 MPa, with a lower deformation of 0.05 mm/m.

Therefore, it was demonstrated that mixing straws from different locations can improve bio-based concrete mechanical characteristics without compromising thermal conductivity (0.078 W.m^−1^.K^−1^). This blend combined the rigidity of Somme straws with the flexibility of Marne straws, allowing it to be classified as an insulating and self-supporting building material.

## 4. Conclusions

This work was carried out in collaboration with CODEM and its tools. This paper investigates the influence of different types of rapeseed straw and their characteristics, such as chemical, physical and microstructural properties on the compressive strength and thermal conductivity of rapeseed straw concrete.

More specifically, it evaluated the influence of climate, harvest year, aggregate size straw and grinding type of rapeseed straw. It was verified that:The year of harvest had no influence on the thermal and mechanical properties of rapeseed concrete, while climate, aggregate size, and type of crushing had a significant effect on the concrete’s compressive strength but little influence on its thermal conductivity;Aggregates of 20 mm provided better compressive strength to rapeseed straw concrete rather than 10 mm since the 20 mm straws prevented the propagation of cracks more efficiently. In contrast, smaller straws improved the thermal performance of the concrete very slightly;The diversification of straw sources led to better mechanical performance for the rapeseed concrete because it combined the hardness of Somme straws with the flexibility of Marne. This variability improved compressive strength by 10% while maintaining good insulating properties;The type of straw grinding had little influence on the performance of the rapeseed concrete. Impact grinding improved mechanical strength slightly but had little effect on thermal performance. This is due to the partial degeneration of the pith and hence of the most porous portion of the straw.

Therefore, where the crop is grown and how soon the crop is collected after harvesting can influence the quality of the straw used in concrete. Additionally, even though straw is not intended to be stored for several years, it must also be protected from moisture. 

## Figures and Tables

**Figure 1 materials-15-08611-f001:**
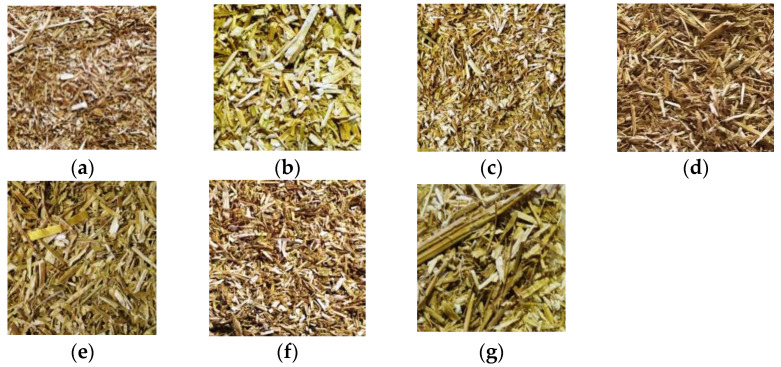
Rapeseed straws from: (**a**)—Marne 15-10, (**b**)—Marne15-20, (**c**)—Marne 16-10, (**d**)—Marne 16-20, (**e**)—Somme 16-10, (**f**)—Somme 16-20, (**g**)—Marne 16-20 Impact.

**Figure 3 materials-15-08611-f003:**
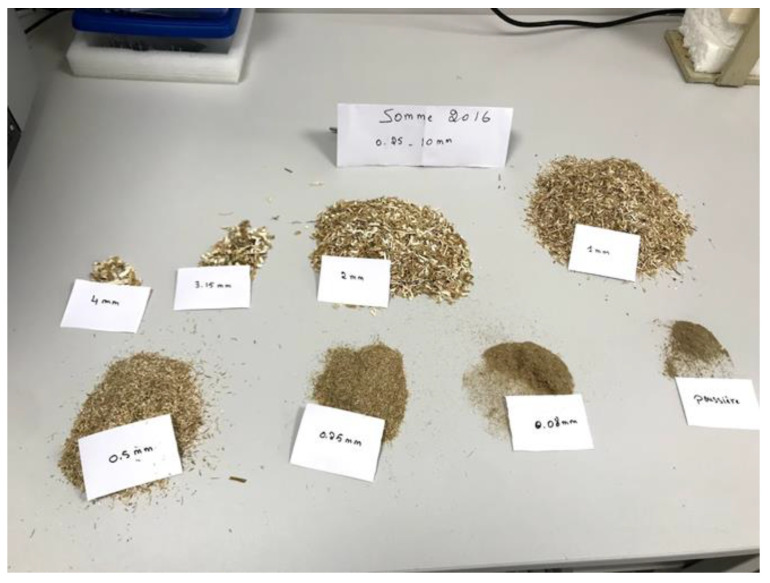
The different sizes of rapeseed straw after the sieving test.

**Figure 4 materials-15-08611-f004:**
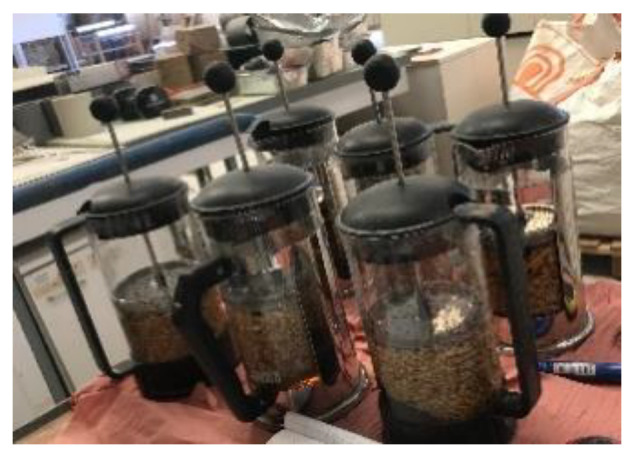
Experimental method for measuring water absorption.

**Figure 5 materials-15-08611-f005:**
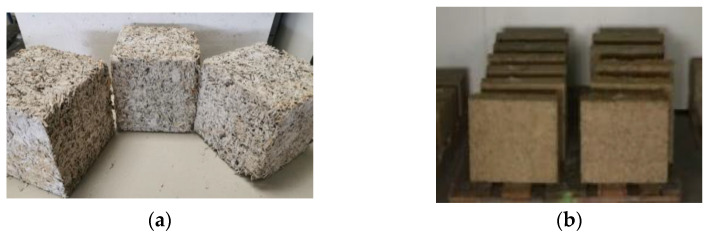
Rapeseed concrete samples for (**a**) mechanical test; (**b**) thermal conductivity test.

**Figure 6 materials-15-08611-f006:**
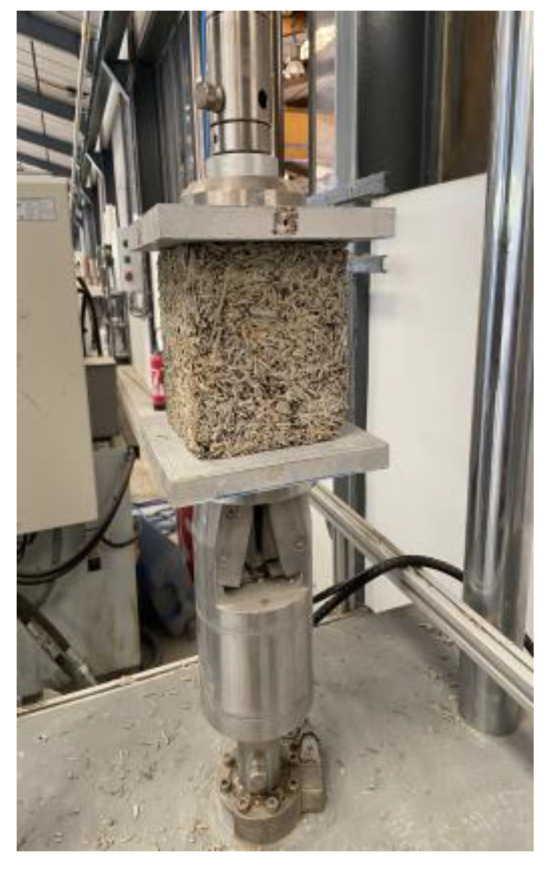
Compression strength test on rapeseed concrete.

**Figure 7 materials-15-08611-f007:**
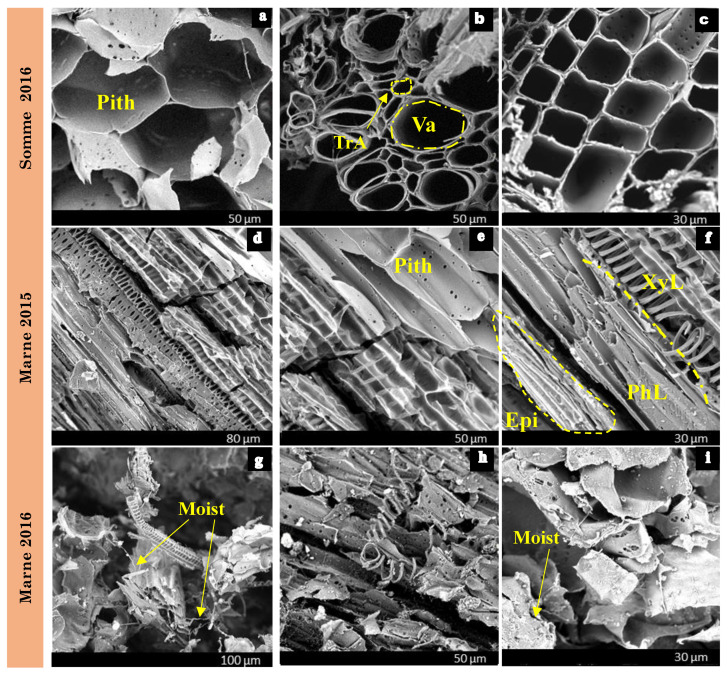
SEM micrographs of rape straws of Somme 2016 (**a**–**c**), Marne 2015 (**d**–**f**) and Marne 2016 (**g**–**i**).

**Figure 8 materials-15-08611-f008:**
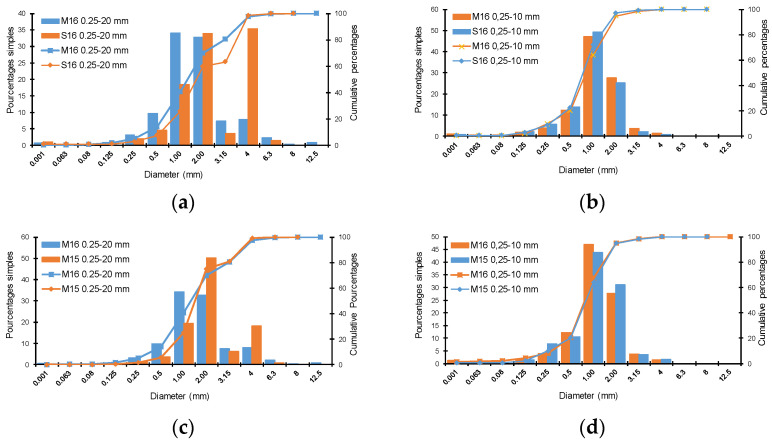
Mechanical sieving results for rapeseed straws: (**a**,**b**)—the climates of Marne and Somme, (**c**,**d**)—the years of harvest, (**e**,**f**)—the size of aggregates, (**g**,**h**)—the type of gridding.

**Figure 9 materials-15-08611-f009:**
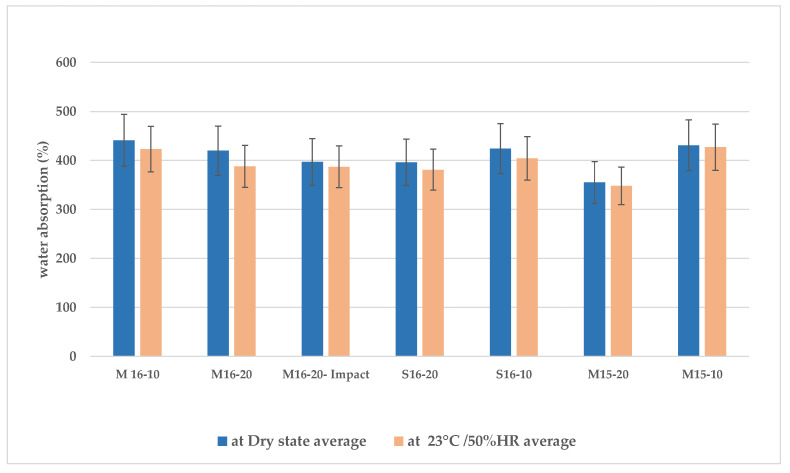
Water absorption of rapeseed straw at 1 h at dry state and 23 °C/50%RH.

**Figure 10 materials-15-08611-f010:**
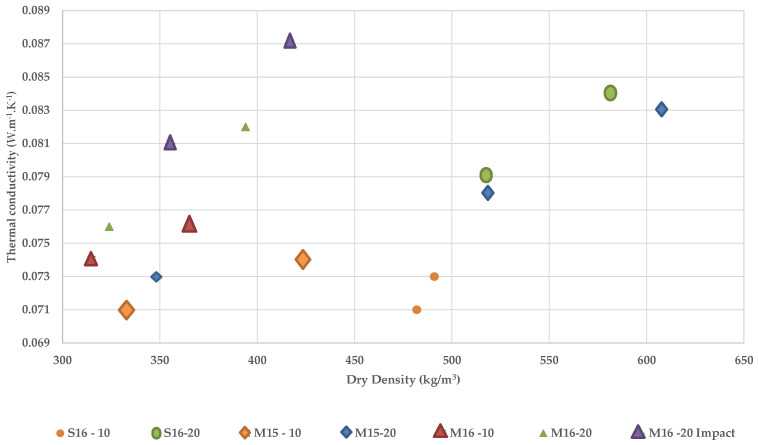
Thermal conductivity as a function of concrete density according to the type of rapeseed aggregates.

**Figure 11 materials-15-08611-f011:**
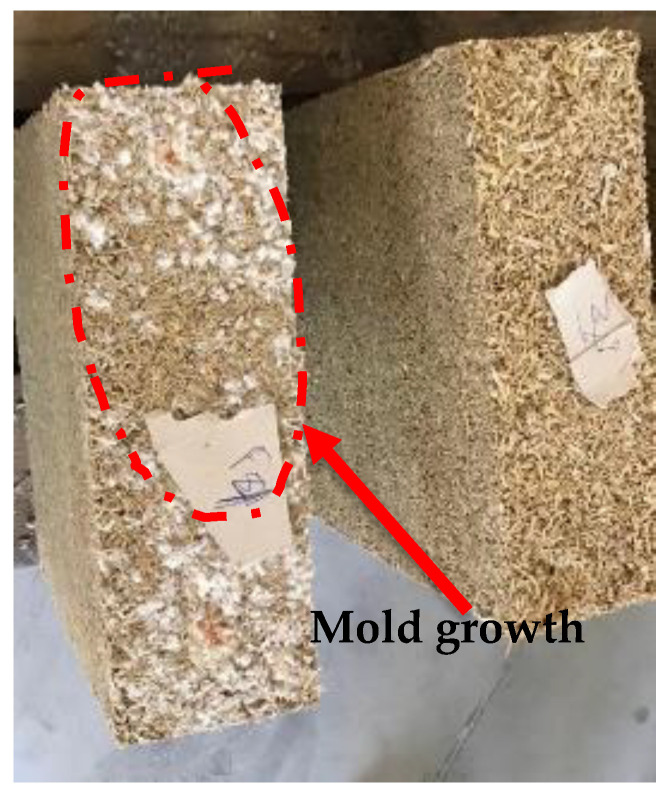
Rapeseed concrete samples made with the M-16 aggregates.

**Figure 12 materials-15-08611-f012:**
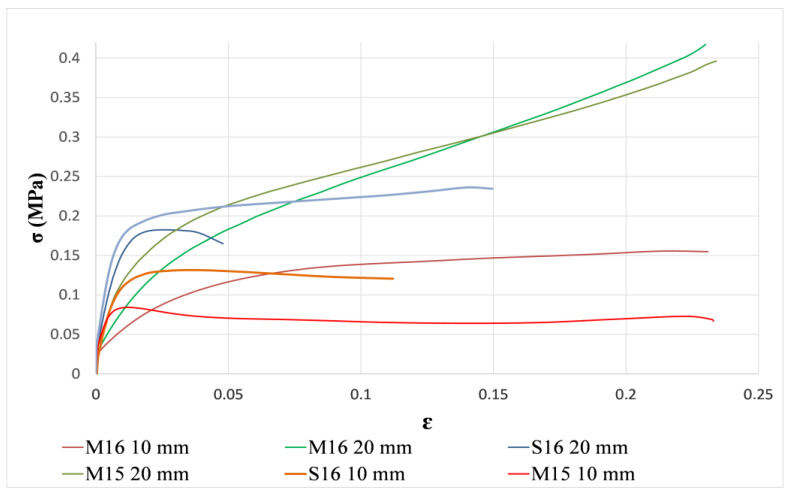
Compressive strength curves of bio-based concrete made from rapeseed plants for different years, harvest locations, and straw sizes.

**Figure 13 materials-15-08611-f013:**
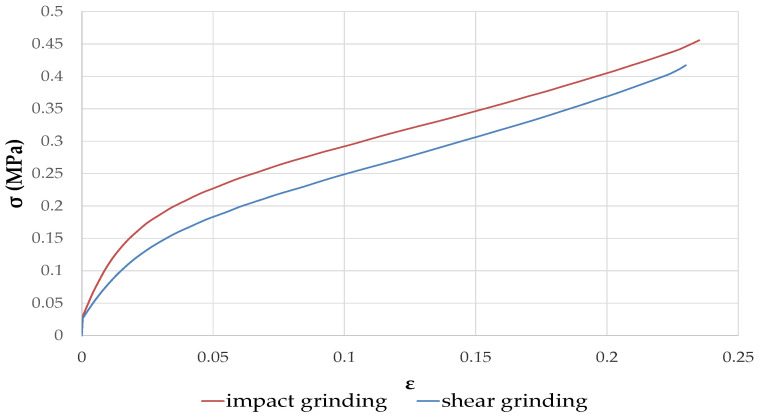
Compressive strength of BINP rapeseed concrete using two grinding techniques.

**Figure 14 materials-15-08611-f014:**
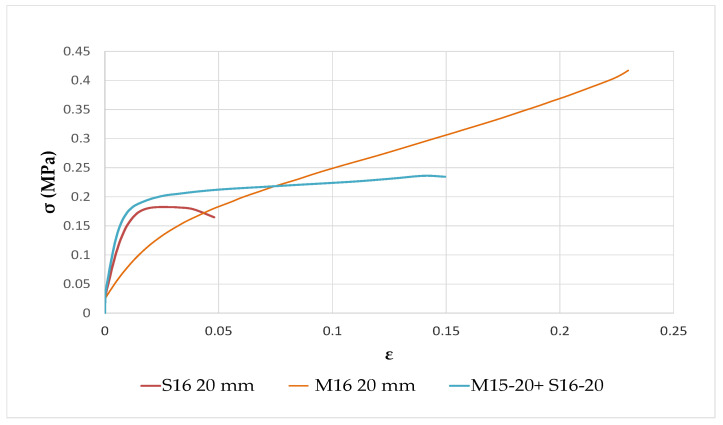
Comparison of the mechanical properties of rapeseed concretes made with M16, S16 straws, and a mixture of both.

**Table 1 materials-15-08611-t001:** Rapeseed straw size according to harvested region.

	M15	M16	S16
0.25–10 mm	X	X	X
0.25–20 mm	X	X	X
0.25–20 mm Impact		X	

**Table 2 materials-15-08611-t002:** Chemical compositions of different aggregate sizes reported in the literature and rapeseed straws investigated.

	Cellulose (%)	Hemicellulose (%)	Lignin (%)	Soluble (%)	Inorganic Materials (%)	
Flax	68–85	10–17	3–5	-	1–2	[47]
Hemp	51.6	21.5	12.9	12.9	6.6	[48]
Hemp	68	9	4.1	-	-	[49]
Hemp	49.97	21.42	9.52	17.75	0.67	[38]
Rape straw	53.06	18.13	9.63	17.68	-	[38]
Rape straw	37.55	31.37	21.30	3.76	6.02	[50]
M 16	51.40	9.30	8.40	29.90	0.90	
S 16	53.20	15.00	10.5	20.90	0.40	
M 15	55.20	12.10	10.90	21.40	0.40	

**Table 3 materials-15-08611-t003:** Particle size distribution of rapeseed straws.

Origin	Dx (10) mm	Dx (50) mm	Dx (90) mm	Dust Content %
M16-10	0.87	1.32	4.38	4.1
M16-20	0.433	1.22	2.38	1.7
M16-20-Impact	0.56	1.54	3.79	1.9
S16-20	0.86	2.25	4.75	0.9
S16-10	0.41	1.16	2.26	2.9
M15-20	0.92	2.04	4.349	0.4
M15-10	0.56	1.30	2.42	2.1

**Table 4 materials-15-08611-t004:** Bulk density of rapeseed straws.

Origin	Density (kg/m^3^)	
	Average	SD
M16-10	90.07	0.55
M16-20	70.24	2.90
M16-20-Impact	66.35	2.87
S16-10	103.95	1.85
S16-20	75.84	1.12
M15-10	101.65	0.87
M15-20	69.29	2.52

**Table 5 materials-15-08611-t005:** Bulk density (ρ_0_), real density (ρs), porosity (n) and length of aggregates from the literature.

	ρ_0_ [kg.m^−3^]	ρ_s_ [kg.m^−3^]	n [%]	Length [mm]	
Hemp shiv	103	1465	92.9	1.25–20	[53]
Hemp shiv	125	1259	90.7	8–15	[30]
Flax shiv	90	1270	92.91	5–15	[10]
Rape straw	130	1162	88.81	10–50	[30]
Hemp shiv	130–87.89	1399	93.72	0–14	[38]
Flax shiv	140–109.91	1321	90.17	0–14	
Rape straw	100–78.71	1385	94.32	7–14	
Rape straw	65	-	94	-	[54]
Rape straw	65	-	-	0–35	[55]

**Table 6 materials-15-08611-t006:** Moisture content of different rapeseed straws.

Origin	Moisture Content %	
	Average	SD
M16-10	13.01	0.23
M16-20	13.95	0.22
M16-20-Impact	13.45	0.42
S16-20	5.23	0.88
S16-10	5.96	0.19
M15-20	5.80	0.33
M15-10	6.01	0.47

**Table 7 materials-15-08611-t007:** Thermal conductivity variation according to the density and type of rapeseed concrete.

Type of Aggregate	Dry Density (Kg/m^3^)	Thermal Conductivity (W.m^−1^.K^−1^)
M15-20	608	0.083
	518	0.078
	348	0.073
M16-20	394	0.082
	324	0.076
S16-20	582	0.084
	530	0.078
	518	0.077
M15-10	502	0.074
	333	0.071
M16-10	365	0.076
	315	0.074
S16-10	491	0.073
	482	0.071
M16-20 Impact	417	0.087
	356	0.081
M15-20 + S16-20	352	0.078

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
