# Peer review of "Physical and Mechanical Properties of Rapeseed Straw Concrete"

_materials, 2022, doi:10.3390/ma15238611_

Round 1

Reviewer 1 Report

Comments are listed below:

1. The authors would explain clearly in the abstract what is the novelty of the proposed method and what is the added value in this article.

2. Strengthen the abstract section. Add the key conclusion of the works in the last two lines of the abstract section. Remove the unnecessary information.

3. Discuss the novelty of the work in respect of the application  

4. There are numerous spelling and grammatical errors. Please revise the manuscript thoroughly. Sentences are also not complete and references are also cited in a rough manner.

5. Please remake the novelty of the text more strongly compared to previous approaches.

6. Try to make a bridge between current and previously published work and specify the gap area and objective of the work. The introduction section is very poor: refer to following published work and write this paragraph in introduction section:

“Wenchen et al. show the creep of aged concrete under extremely high sustained load. They also investigate the behavior of transverse reinforcement on the reinforced concrete creep during the loading and sustained load process. The data in this paper is precious because almost no relevant experiments can be found in previous research. The conclusion made in this research, such as a higher transverse reinforcement ratio can increase axial load capacity and reduce the concrete, especially in the early loading stage, provide great help for this research.”

- Ma, Wenchen, Ying Tian, Hailong Zhao, and Sarah L. Orton. "Time-Dependent Behavior of Reinforced Concrete Columns Subjected to High Sustained Loads." Journal of Structural Engineering 148, no. 10 (2022): 04022161.

7. The proposed methodology, contains many well know equations, which are already published. The methodology sections should be reduced and appropriate references should be added. The authors should keep only their new contributions.

8. in some sentences there is an excess if use of the preposition "of".

9. The results are ok but the discussion section is very poor. It looks like a technical report instead of a technical article. Improve the discussion section and add more references in support of the results.

10. Shorten the length of the conclusion section. Please focus on the main achievements of the approach.

11. The work is good, but the technical discussion and introduction section needs improvement. Paper can be accepted after following minor corrections.

12. The author should check typing errors throughout the manuscript. English style should also be improved.

For these reasons, I recommend the acceptance of this manuscript for publication after Major Revision.

Author Response

Dear Reviewer,

First of all, thank you for your effort to improve the quality and the comprehension of the paper entitled “Physical and mechanical properties of rapeseed straw concrete."

Please find below an item-by-item response to the reviewers' comments concerning the manuscript. Your initial remarks are mentioned in italics, and the correction is explained in red. Also, find the revised manuscript with all additions/changes marked up using the "Track Changes" function.

We hope that these modifications will allow the publication of the article under the best conditions.

Thank you in advance for your time and attention.

Reviewer 2 Report

This research examines a novel building material comprised of rapeseed concrete. This material is a non-load-bearing insulating concrete (BINP) developed for use in wood-frame wall construction. It is composed of rape straw, lime, and cement particles. I recommend for publication after considering the following minor points:

1. The abstract should add the potential of this research in real life as well as in engineering practice.

2. The authors have written a very logical and convincing introduction. However, this study is experimental, so to enrich this section, some numerical investigation, which is related to beam and slab structures that are widely used in construction, can be added as follows: "Bending of Symmetric Sandwich FGM Beams with Shear Connectors," "Numerical Investigation on Static Bending and Free Vibration Responses of Two-Layer Variable Thickness Plates with Shear Connectors," "Static bending analysis of symmetrical three-layer FGM beam with shear connectors under static load," "The Third-Order Shear Deformation Theory for Modeling the Static Bending and Dynamic Responses of Piezoelectric Bidirectional Functionally Graded Plates," "A new efficient modified first-order shear model for static bending and vibration behaviors of two-layer composite plate."

3. As depicted in Table 1 and Figure 1, is rape straw size based on a technical standard?

4. Figures 3 and 5b should be improved to be easy to observe for readers.

5. Equation 1 should be cited from value references.

6. Are the dimensions of specimens (rapeseed concrete samples) based on a technical standard? This should be cited.

7. The labels of the horizontal and vertical axes in Figure 8 need to be returned to English.

8. The results discussed by the authors are very rich and interesting.

9. Some minor errors in English were found; the author should revise the entire article.

In general, I thought the paper did an outstanding job of presenting its methodology, its findings, and its subsequent commentary. When natural materials are investigated and implemented in life and engineering in response to the various environmental concerns that exist all over the world, I believe that the study results presented in this article will be appealing to readers.

Author Response

(The authors gave the same response as above.)

Reviewer 3 Report

The study “Physical and mechanical properties of rapeseed straw concrete ” is fine and novel, the results are comprehensive, the discussion is deep. I think it can be accepted for publication after a minor revision. My comments are as follow.

1. The introduction part should be more comprehensive. Why did the authors produce non-load-bearing concrete? What is the application background of this concrete, since the 0.22MPa is very low. Besides, the authors should pay attention not only to a new materials but also to the engineering background and its potential use.

2. Why was the thermal conductivity of concrete the major property the authors studied?

3. The authors claimed that “one of the goals is to reduce the cost of materials while maintaining their durability”. What are the durability properties of this concrete?

4. The pictures in Fig.3-5 are not clear, please replace them by some clear ones with at least 300 dpi.

5. What are the mix proportions of concretes in this study.

6. How about the test procedure of concrete strength. How many samples used in strength tests.

7. The compressive strength results are values rather than some curves. Giving the definite strength values are enough.

Author Response

(The authors gave the same response as above.)

Reviewer 4 Report

The manuscript, materials-2028429, reported their research on the preparation and characterization of innovative building material based on rapeseed concrete. Over all, this topic is interesting and important. A lot of data was provided and discussed, however, the data needs more deeply discussed. This work may be interested in engineers in field of biomass materials and building materials. Followings are some comments for the manuscript,

1. Considering about the microarchitectures of the aggregates of Figure 7, is the mechanical properties tested directional? I mean how difference between the data tested from vertical direction and horizontal direction?

2. In addition, a small thing that the “b” and “c” were missing in Figure 7.

Author Response

Dear Reviewers,

First of all, thank you for your effort to improve the quality and the comprehension of the paper entitled “Physical and mechanical properties of rapeseed straw concrete."

Please find below an item-by-item response to the reviewers' comments concerning the manuscript. Your initial remarks are mentioned in italics, and the correction is explained in red. Also, find the revised manuscript with all additions/changes marked up using the "Track Changes" function.

We hope that these modifications will allow the publication of the article under the best conditions.

Thank you in advance for your time and attention.

Best regards,

Reviewer 5 Report

The manuscript presents the results of an investigation conducted to examine the compressive strength and thermal conductivity of rapeseed straw concrete.

Reducing the environmental impact of the construction sector and increasing the energy efficiency of the built environment is a relevant scientific topic for which the development of new building materials with improved thermal properties, produced from locally available secondary resources, is of utmost importance. The work described in the manuscript is well aligned with this research topic and therefore suitable for the journal.

However, authors are encouraged to conduct a profound revision of the manuscript and make additional efforts to:

However, authors are encouraged to conduct a profound review of the manuscript before publication and make additional efforts to:

i)      highlight the novelty of their work compared to other research works that has examined rapeseed concrete.

ii)     Thoroughly characterize all raw materials used, define the mix design of each formulation and extend the experimental program beyond just two properties of the concrete specimens

iii)   Improve the structure and scientific soundness of the manuscript. Sections that should belong to "Introduction" and "Materials and methods" are often intermingled with results

iv)     Resume the "Materials and methods" section to essential and restrict discussion of methods to the appropriate section.

v)      Eliminate all non-relevant and redundant figures. All captions should be provided in English.

vi)     Strengthen the analysis and increase the depth and clarity of the discussion. All materials and design decision should be carefully justified and discussed.

vii)   Provide data to support all claims made. Most certainly is hardly acceptable in the scientific literature.

viii)  Proofread the manuscript, use tense paste, avoid the use of colloquial expressions and rephrase whenever necessary to increase the overall quality of the text and English language.

Author Response

(The authors gave the same response as above.)

Round 2

Reviewer 2 Report

The authors have clearly explained the issues pointed out in round 1.

Reviewer 3 Report

It can be accepted

Reviewer 4 Report

The quality of present manuscript (materials-2028429) has been significantly improved according to the suggestions of reviewers.